# Epidemiological, Clinical, and Genomic Profile in Head and Neck Cancer Patients and Their Families

**DOI:** 10.3390/biomedicines10123278

**Published:** 2022-12-17

**Authors:** Thiago Celestino Chulam, Fernanda Bernardi Bertonha, Rolando André Rios Villacis, João Gonçalves Filho, Luiz Paulo Kowalski, Silvia Regina Rogatto

**Affiliations:** 1Department of Head and Neck Surgery and Otorhinolaryngology, A.C. Camargo Cancer Center, São Paulo 01509-001, SP, Brazil; 2Department of Pediatrics, Faculdade de Medicina da Universidade de São Paulo (FMUSP), São Paulo 01246-903, SP, Brazil; 3Department of Genetics and Morphology, Institute of Biological Sciences, University of Brasília (UnB), Brasília 70910-900, DF, Brazil; 4Department of Clinical Genetics, University Hospital of Southern Denmark, Beriderbakken 4, 7100 Vejle, Denmark; 5Institute of Regional Health Research, Faculty of Health Sciences, University of Southern Denmark, 5000 Odense, Denmark

**Keywords:** head and neck cancer, familial cancer, cancer predisposition, risk factors, copy number alterations

## Abstract

Inherited cancer predisposition genes are described as risk factors in head and neck cancer (HNC) families. To explore the clinical and epidemiological data and their association with a family history of cancer, we recruited 74 patients and 164 relatives affected by cancer. The germline copy number alterations were evaluated in 18 patients using array comparative genomic hybridization. Two or more first-degree relatives with HNC, tobacco-associated tumor sites (lung, esophagus, and pancreas), or other related tumors (breast, colon, kidney, bladder, cervix, stomach carcinomas, and melanoma) were reported in 74 families. Ten index patients had no exposure to any known risk factors. Family members presented tumors of 19 topographies (30 head and neck, 26 breast, 21 colon). In first-degree relatives, siblings were frequently affected by cancer (*n* = 58, 13 had HNC). Breast cancer (*n* = 21), HNC (*n* = 19), and uterine carcinoma (*n* = 15) were commonly found in first-degree relatives and HNC in second-degree relatives (*n* = 11). Nineteen germline genomic imbalances were detected in 13 patients; three presented gains of WRD genes. The number of HNC patients, the degree of kinship, and the tumor types detected in each relative support the role of heredity in these families. Germline alterations may potentially contribute to cancer development.

## 1. Introduction

Head and neck cancer (HNC) is the seventh most common tumor type worldwide, comprising a heterogeneous group of upper aerodigestive tract malignancies [1]. Most cases (90%) are histologically classified as squamous cell carcinoma. Despite several therapeutic advances in cancer treatment, patients with advanced-stage disease presented a 5-year survival rate varying from ~20 to 60% [2]. The most significant improvements in survival were described only among patients with nasopharynx and oropharynx cancer [3,4].

The main risk factors associated with HNC development are tobacco usage, alcohol consumption, and human papillomavirus (HPV) infection [5,6,7,8]. Rodriguez et al. [9] reported a 20-fold increase in the risk of developing oral and pharyngeal carcinomas in individuals younger than 46 years and smokers and a five-fold increased risk for alcohol consumers. According to the authors, an increased risk of almost 50 times was found in consumers of both alcohol and tobacco [9]. However, chronic exposure to alcohol and tobacco does not adequately explain the presence of a family history of HNC [10]. A hereditary factor associated with HNC has been suggested by the presence of the disease in first-degree relatives, in patients at early onset not exposed to known risk factors, and in families with a large number of relatives affected by cancer [11].

Epidemiological studies using a large case-control cohort and controlling differences in age, sex, race, tobacco use, and alcohol consumption, revealed an increased risk of patients developing HNC in families [12,13,14,15,16,17,18]. A significantly increased risk of oropharyngeal cancer was reported in the first-degree relatives of HNC patients and those with multiple tumor types, including lung and cervical cancers [19]. Overall, these studies pointed out an increased risk of approximately two to three-fold in individuals with a family history of head and neck cancer.

We previously described the *PHF21B* deletion, loss of function, and changes in DNA methylation in a cohort of HNC patients, including 40 cases with a family history of cancer [20]. Pathogenic germline variants in *CDKN2A*, *RECQL4*, and *SDHB* were described as associated with a high risk of oral carcinomas in young patients [21,22]. Recently, we described an association between the germline variants in DNA repair genes with an increased risk of HNC development in young patients [23]. We also found that germline variants in DNA repair genes could improve survival, while the *FAT1* germline alterations were associated with worse survival [23].

Several studies have described the genomic alterations in HNC [24,25]. The copy number alterations (CNAs), including losses of 3p and 8p, and gains of 3q, 5p, and 8q, have been reported in head and neck tumors. Although the germline genomic imbalances have been previously associated with the susceptibility of a series of human diseases, including cancer, the germline CNAs in individuals with HNC and a family history of cancer, are largely unknown [26].

In this study, we performed an epidemiological and clinical evaluation of 74 HNC families in which the index case had HNC, and their relatives presented HNC or related tumors. In a subset of 18 HNC patients, we evaluated germline CNAs using array-CGH (comparative genomic hybridization). We sought to better understand the phenotypic presentation and the variables involved in the risk of developing HNC in families.

## 2. Materials and Methods

### 2.1. Study Participants

This study comprised 74 patients presenting a family history of cancer (and alive at the time of the data collection) selected at the hospital-based cancer registry of the Head and Neck Surgery and Otorhinolaryngology Department at the A.C. Camargo Cancer Center, São Paulo, Brazil, from 2003 to 2011. The patients and relatives were selected during regular follow-up consultations and received genetic counseling. The clinical and pathological data and medical history from all patients were retrieved from the medical records. The tumor site and histopathological data from the relatives were recorded (when available). When possible, the evidence of cancer was based on the assessment of the medical records or ascertained from the death certificate. The HPV genotyping was evaluated using the Linear Array HPV Genotyping Test (Roche Molecular Systems, Alameda, CA, USA). The array-CGH was performed in 18 of 74 HNC patients (blood samples) and in the sister (27.2) with colorectal cancer of one index patient (27.1). All patients provided written informed consent. The Institutional Human Research Ethics Committee approved this study (CEP #792/2006B October 31, 2008 and CEP #1200/09 May 4, 2009).

The criteria used to characterize the high risk HNC families were previously described in [20]. In short, only patients with head and neck squamous cell carcinomas (excluding tumors located at the nasopharynx, thyroid, and salivary glands) with complete clinical and pathological data were included. Patients clinically diagnosed with Fanconi’s anemia, xeroderma pigmentosum, epidermolysis bullosa, and juvenile papillomatosis were excluded. Three major inclusion criteria were applied: the presence of at least two first-degree relatives affected with HNC and or other related cancer (breast, colon, kidney, bladder, cervix, stomach carcinomas, and melanoma) and/or tobacco-associated tumors (lung, esophagus, pancreas, and head and neck) in the same family; a history of cancer in patients younger than 45 years old in at least one relative; any age at onset in no tobacco and or alcohol consumers. The family history of cancer considered first and second-degree relatives with cancer as informative, excluding any foster or step-relatives with cancer.

We collected information on the number of patients and relatives with HNC, the presence of the related tumors or tobacco associated-tumors, the primary sites of the tumors, the demographic characteristics (such as age, gender, race, birth year, place of birth, family income, and education), occupational exposure, tobacco usage, alcohol consumption, frequency of visits to the dentist, and the use of a dental prosthesis.

### 2.2. DNA Extraction and Array-CGH and the Data Analysis

The genomic DNA was isolated using sodium dodecyl sulfate/proteinase K digestion followed by the phenol-chloroform extraction and ethanol precipitation. High-quality genomic DNA (200–500 ng) from the cases and a reference (paired male/female commercial genomic DNA) (Promega, Madison, WI, USA) were hybridized on Human Genome CGH Microarray 4 × 180K (Agilent Technologies, Santa Clara, CA, USA), according to the manufacturer’s instructions. The array data were extracted using the default CGH settings of the Feature Extraction software v10.7 (Agilent Technologies, Santa Clara, CA, USA), and analyzed with the CytoGenomics software v5.2.1.4 (Agilent Technologies, Santa Clara, CA, USA). The copy number variations (CNVs) were called, according to the following criteria: algorithm ADM-2, sensitivity threshold at 6.0, fuzzy zero correction, at least four consecutive altered probes, and log2 ratio < −0.25 for losses and >0.25 for gains. All CNVs were compared to the Database of Genomic Variants (DGV, http://dgv.tcag.ca/dgv/app/home, updated on 25 February 2020, and accessed on 20 September 2022) and a database composed of 100 healthy Brazilian individuals [27]. Only rare CNVs (≤5% of the Brazilian reference population and ≤0.05% of the DGV database) were considered in this study.

## 3. Results

The median age of the index patients was 60 years (ranging from 28 to 78), and only three patients were younger than 45. The disease was predominantly detected in the male gender (75%). At least one first-degree relative affected by cancer was recorded. Sixty-three (85%) patients were Caucasian, 62% were born in urban areas, and 19% had completed higher education. Sixty patients (81%) were smokers, and 59 (79.7%) consumed alcoholic beverages. The most common tumor site associated with familial cancer was the oral cavity (31 cases), followed by the larynx (24 cases). Five (6.7%) individuals were genotyped as HPV16 (one oral cavity and four oropharyngeal carcinomas). The epidemiological data are detailed in Table 1.

A total of 164 relatives were affected by cancer (117 first-degree and 47 second and third-degree relatives) from 19 different topographies (29 head and neck, 28 breast, 21 colorectal, 18 stomach, and 18 cervix carcinomas). The most common tumor type in first-degree relatives was breast carcinoma (21 cases), followed by head and neck (19 cases), uterus (15 cases), colorectal (14 cases), and gastric carcinomas (12 cases). The most frequent tumors found in second-degree relatives were head and neck (11 cases), colon (7 cases), breast (5 cases), and gastric carcinomas (5 cases) (Table 2).

In first-degree relatives, siblings were the most frequently affected by cancer (58 cases), followed by the father (31 cases) and mother (28 cases). Four relatives presented cancer during childhood. Fathers presented more frequently with stomach cancer (7 cases), while mothers had uterine cervix and breast carcinomas (7 cases each). Breast carcinomas (14 cases) followed by head and neck cancer (13 cases) were the most frequent tumor types found among siblings (Table 2).

The index cases with oral cancer were more frequently associated with brothers with head and neck cancer (7/18 cases), while in oropharyngeal cancer index patients, siblings remained the most affected group, with breast cancer as the most common associated tumor (3/9 cases). The index patients with larynx cancer presented a significant predominance of siblings affected by cancer, compared to fathers and mothers (22, 9, and 7 cases, respectively). In this sample set, an association with breast and head and neck tumors was also observed (six and five cases, respectively).

Among the first-degree relatives and independent of sex, siblings were the most frequently first-degree relative affected by cancer (mainly breast and head and neck carcinomas), followed by fathers and mothers. However, the tumor type analysis in each affected family revealed a distinct pattern. In male patients, a predominance of siblings (44 cases) was detected, while in female patients, siblings and parents were equally affected (10 cases each). In male index cases, a colorectal tumor was the most common tumor type in fathers, while in mothers, it was breast carcinoma. A predominance of breast and head and neck cancer was observed in brothers/sisters from male index cases (14 and 12, respectively). The father and mother/sister of female index patients presented more frequently with stomach and cervix uterine cancer, respectively (Table 2). Appendix A presents the detailed information on the index patients evaluated by array-CGH. A representative pedigree of one family investigated in our study is shown in Figure 1A.

Ten non-smokers and non-alcoholic patients (aged 28 to 78 years old) were detected among our 74 index cases; six of them presented advanced clinical stages. Seven of these ten probands presented oral cavity tumors, and each of their mothers were affected by cancer.

### Copy Number Variations

Five (61.1, 65.1, 67.1, 72.1, and 74.1) out of 18 patients analyzed had no CNVs (Table 3). The remaining 13 showed up to two germline CNVs each, with a total of 12 gains and seven losses. The 19 rare germline genomic imbalances encompassed more than 100 genes, including a known oncogene (*TPR*) and two tumor suppressor genes (*N4BP2* and *RHOH*). The rare CNVs in chromosomes 2 and 19 were identified in three different patients. The index case 27.1 (56 years old, oral cavity tumor) and her sister (27.2, 58 years old, colorectal cancer) shared the same deletion on chromosome 5p15.2, covering an intronic region of the *LINC01194* gene. Additionally, the sister also presented a rare gain of chromosome 11q13.2, encompassing the *RHOD* and *KDM2A* genes (Table 3). Seven index patients presented rare CNVs covering miRNAs and/or lncRNAs (Table 3). Of note, three patients (51.1, 207, and 339) older than 60 years, showed gains covering genes belonging to the WD-repeat (WDR) gene family (Figure 1B and Table 3).

## 4. Discussion

The role of hereditary factors in head and neck tumors has been suggested, based on the clinical reports [19,28,29] and case-control studies [12,13,14,16]. Furthermore, the environmental factors shared between individuals of the same family do not explain the number of siblings and other first-degree relatives affected by cancer [12,13,18,19]. In our study, we investigate the clinical and epidemiological factors in families with head and neck cancer supporting the role of the genetic predisposition associated with the risk of developing cancer in these families.

A population-based case-control study of oral and oropharyngeal cancer showed a slightly increased risk of these tumors associated with a family history of cancer [16]. A higher risk of developing HNC was detected when an oral/pharyngeal, larynx/esophagus, or lung cancer occurred in parents or siblings [16]. In our study, the HNC risk was more closely related to the oral cavity and larynx tumors. In addition, siblings were more frequently affected by cancer in our families.

Alcohol and tobacco are well-established risk factors associated with head and neck cancer. Approximately 75% of HNC patients are alcohol and tobacco consumers [8]. In the present study, 14% of the patients were non-users of tobacco or alcohol, 20% were non-smokers, and 20% had no history of alcohol consumption. The occasional use of tobacco and alcohol was higher, with up to 30% of the patients reporting low tobacco consumption and social drinking habits. According to Negri et al. [17], family history is even more important when individuals are exposed to other concomitant risk factors.

Human papillomavirus infection has been described as a risk factor for HNC, especially in oropharyngeal carcinomas. HPV16 is the main oncogenic subtype related to oropharynx neoplasms, accounting for ~83% of HPV-positive cases [30]. A recent study of 254 oropharyngeal carcinomas from Brazilian patients reported a prevalence of ~32% of HPV16 [31]. We detected only five patients (6.7%) genotyped as HPV16 in our cohort (four in the oropharynx and one in the oral cavity). These findings suggested that the family history of cancer reported in our patients has more impact on the cancer risk than the HPV infection.

Based on the paucity of evidence and the etiological factors involved in the risk of the disease, such as tobacco and alcohol consumption, the familial HNC syndrome concept is poorly described in the literature. The lack of supporting or contradictory evidence confirms the unfamiliarity and novelty of this clinical entity. The clinical suspicion for a familial HNC syndrome is low and obtaining an accurate and detailed family history has not been a priority in clinical practice. However, these data have the potential to benefit patients through effective genetic counseling and early intervention [21].

We also investigated DNA copy number variations as a risk factor in 18 index patients from head and neck cancer families. The germline variants have been associated with HNC, such as *CDKN2A* (cyclin-dependent kinase inhibitor 2A), *FAMMM* (familial atypical multiple mole melanoma syndrome), and *ATR* (ataxia telangiectasia and Rad3-related) genes, which are involved with oropharynx cancer in families presenting with skin, breast, and cervical cancers. Fanconi anemia genes (congenital abnormalities, bone marrow failure, and a genetic predisposition to leukemia and squamous cell carcinomas) have been related to a high risk of developing head and neck cancer [32]. The high prevalence of cancer in young patients is a feature associated with hereditary cancer syndromes. Recent studies described the germline variants of *CDKN2A*, *RECQL4*, and *SDHB* in the DNA repair genes associated with a high risk of head and neck cancer in young patients [21,22,23]. Previously, we described the *PHF21B* deletion, loss of function, and changes in DNA methylation in HNC patients with a family history of cancer [20].

We found rare germline CNVs in 13 of 18 unrelated index patients. No alterations were detected, compared to the gene list of head and neck cancer-associated genes (Appendix A from reference [23]). Interestingly, three patients with oral cavity tumors (51.1, 207, and 339) presented gains encompassing WDR genes (*WDR83*, *WDR19*, and *WDR47*, respectively). WDR is one of the largest gene families in eukaryotes, whose proteins are involved in the protein transport, chromatin modification, and signal transduction [33]. WDR proteins have been associated with cancer development [34]. Specifically, *WDR34* was suggested as a potential tumor suppressor gene in oral cancer [35]. The upregulation of *WDR83* and *WDR19* has been implicated in gastric and prostate cancers, respectively [36,37]. Germline alterations have also been described as predictive of outcomes in cancer patients. Chatrath et al. [38] described 79 germline variants in individual cancers and 112 prognostic germline variants in groups of cancer. Among the genes, the authors reported *WDR36* (rs7705304) as a prognostic germline variant in colon adenocarcinoma [38]. Overall, germline alterations in WDR genes can contribute to developing cancer in these families.

Three patients (2.1, 26.1, and 207) presented gains covering MIR genes (*MIR1256*, *MIR548F1*, and *MIR574*, respectively). MIRs act in the gene regulation and have been previously associated with HNC [39,40]. These three miRNAs (*MIR1256*, *MIR548*, and *MIR574*) have also been related to cancer progression in gastric, thyroid, and brain cancer [41,42,43]. One patient (27.1) and her sister (27.2; colorectal cancer) presented a loss of the long non-coding RNA *LINC01194*. *LINC01194* acted as an oncogene in colorectal cancer and was associated with a poor survival [44]. Previous studies described that the upregulation of this lncRNA enhances the malignant potential of triple negative breast and laryngeal cancers [45,46]. The role of *LINC01194* as a germline predictive marker deserves further investigation. Interestingly, the sister (27.2) of our index patient (27.1) also presented a gain of 11q13.2, encompassing 4 of 5 exons of *RHOD* and 8 of 21 exons of the *KDM2A* (lysine demethylase 2A) gene. KDM2A mainly recognizes the unmethylated region of the CpG islands and demethylates histone H3K36 residues. KDM2A has a role in chromosome remodeling, gene transcription, cell proliferation and differentiation, cell metabolism, and gene stability [47].

We also found some CNVs of particular interest. For patient 26.1, the same gain covering *MIR548F1* also encompassed *TPR* (translocated promoter region, nuclear basket protein), a gene that codifies a member of the nuclear pore complex [48,49]. Another patient (53) showed a gain covering 4 of 10 exons of the *ASMT* gene (acetylserotonin O-methyltransferase), which codifies a key enzyme involved in synthesizing melatonin. Melatonin is chemopreventive with tumor-inhibitory effects in a variety of in vitro and in vivo models of neoplasia [50]. Patient 229 presented a heterozygous loss of the *TICAM1* (TIR domain-containing adaptor molecule 1) and *PLIN3* (lipid droplet-related protein perilipin-3) genes. Perilipins are structural proteins associated with lipophagy and lipid droplet integrity, and their overexpression is associated with tumor aggressiveness, while *TICAM1* participates in immune and inflammation responses to malignant cells [51,52].

Although with a significant clinical impact, our study has limitations. For instance, the death certificates of all relatives of our index cases could not be obtained to confirm the family cancer history. However, the number of individuals affected, the degree of kinship, and the tumor types detected in each relative, support the role of heredity in these families. As previously mentioned, the family history of head and neck cancer is not an isolated risk factor, and nor it is well documented as other tumor subtypes (such as breast and colorectal cancer). However, this study has favored the importance of family history as a supporting factor that should be cherished and subsequently detailed in other studies. Further molecular studies using families with a strong hereditary component have the potential to identify the genes associated with a head and neck cancer predisposition. In addition, our power to detect the predisposition genes was limited due to the number of cases, the need for experimental investigation and segregation analysis, and the use of an array platform. Implementing a different protocol, such as whole genome sequencing, can point out new genes and alterations in selected families using well-established criteria.

## 5. Conclusions

The clinical and epidemiological factors described in our families with head and neck cancer support the role of a genetic predisposition associated with the risk of developing cancer in these families. We also demonstrated that the germline copy number variations could contribute to the cancer risk.

## Figures and Tables

**Figure 1 biomedicines-10-03278-f001:**
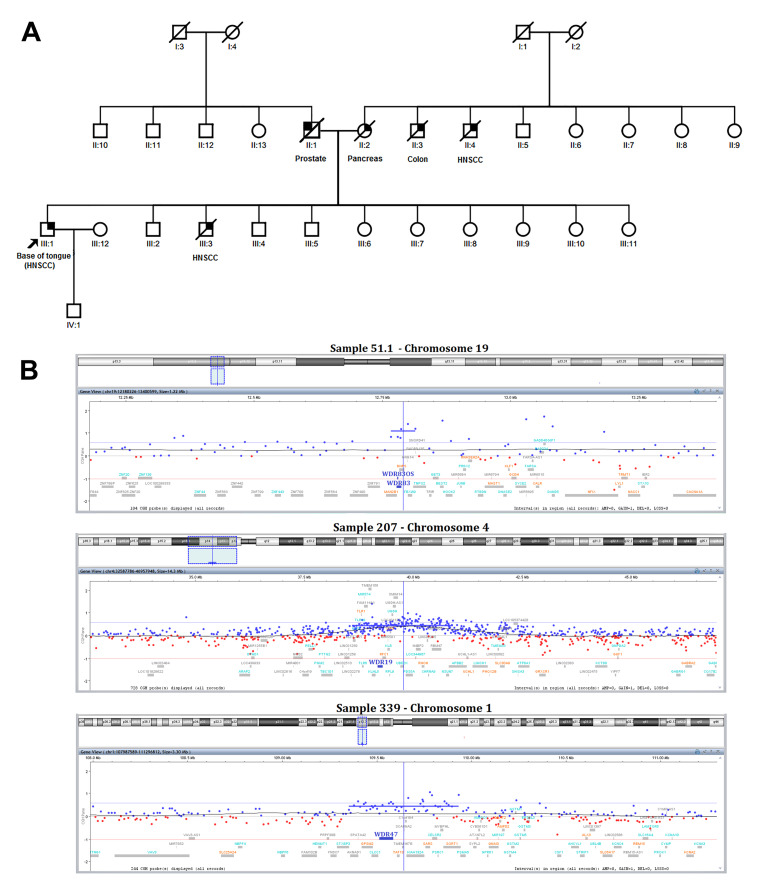
(**A**). Pedigree of one family showing two generations of patients with head and neck cancer (II:4 and III:3) and other tumor types, in relatives of the index patient (III:1). (**B**). Graphical representation of the gains encompassing the genes of the WDR family.

**Table 1 biomedicines-10-03278-t001:** Epidemiological and clinical characteristics of 74 index patients, including 18 evaluated by array-CGH.

Variable	Category	Frequency (%) (*n* = 74)	Array-CGH (*n* = 18)
Gender	Male	56 (75.7)	15
	Female	18 (24.3)	3
Age (years)	≤45	3	1
	>45	71	17
Race	Caucasian	63 (85.2)	18
	Not Caucasian	11 (14.8)	0
Smoking	No	13 (18.9)	4
	Yes	61 (81.1)	14
Passive Smoking	No	63 (82.5)	15
	Yes	13 (17.5)	3
Alcohol Consumption	No	15 (20.3)	4
	Yes	59 (79.7)	14
Tumor Site	Oral Cavity	31 (41.9)	10
	Oropharynx	15 (20.3)	6
	Larynx	24 (32.4)	0
	Hypopharynx	4 (5.4)	2
Clinical Stage	I	12 (16.2)	2
	II	11 (14.9)	3
	III	15 (20.3)	4
	IV	36 (48.6)	9
HPV Status	Yes (HPV16)	5 (6.7)	2
	No	60 (81.1)	14
	NA	9 (12.2)	2
Status	Alive without disease	65 (87.8)	11
	Alive with disease	5 (6.8)	2
	Died by disease or other causes	5 (6.8)	4

NA: not available.

**Table 2 biomedicines-10-03278-t002:** Distribution of the tumors among family members of head and neck cancer index patients.

Tumor	Father *n* = 31 (%)	Mother *n* = 28 (%)	Sibling *n* = 58 (%)	Child *n* = 4 (%)	1st Degree Relatives (%)	2nd Degree Relatives (%)
Head and Neck	5 (16.1)	1 (3.6)	13 (22.4)	-	19 (15.7)	11 (27.5)
Breast	-	7 (25.0)	14 (24.1)	-	21 (17.4)	5 (12.5)
Stomach	7 (22.6)	3 (10.7)	2 (3.4)	1 (25.0)	13 (10.7)	5 (12.5)
Esophagus	2 (6.5)	-	4 (6.9)	-	6 (5.0)	3 (7.5)
Colon	5 (16.1)	4 (14.3)	5 (8.6)	-	14 (11.6)	7 (17.5)
Prostate	5 (16.1)	-	2 (3.4)	-	7 (5.8)	1 (2.5)
Melanoma	1 (3.2)	-	-	-	1 (0.8)	-
Lung	2 (6.5)	3 (10.7)	3 (5.2)	-	8 (6.6)	3 (7.5)
CNS	-	-	-		-	1 (2.5)
Liver and Biliary Tract	2 (6.5)	-	1 (1.7)	-	3 (2.5)	-
Thyroid	-	2 (7.1)	1 (1.7)	-	3 (2.5)	-
Uterine Cervix	-	7 (25.0)	7 (12.1)	1 (25.0)	15 (12.4)	3 (7.5)
Pancreas	-	1 (3.6)	3 (5.2)	-	4 (3.3)	1 (2.5)
Skin	-	-	1 (1.7)	-	1 (0.8)	-
Leukemia	-	-	-	1 (25.0)	1 (0.8)	-
Kidney	-	-	1 (1.7)	-	1 (0.8)	-
Bladder	1 (3.2)	-	1 (1.7)	-	2 (1.7)	-
Testicle	-	-	-	1 (25.0)	1 (0.8)	-

CNS: central nervous system.

**Table 3 biomedicines-10-03278-t003:** Rare germline copy number alterations detected in the blood samples of 14 out of 18 patients.

Sample	Chr	Cytoband	Start	End	#Probes	Event	Log2 Ratio	*p*-Value	miRNAs/lncRNAs	Genes	Note
2.1	19	q12	29,816,969	29,877,356	4	loss	−0.729499	6.89 × 10^−11^	-/*VSTM2B-DT*	-	Only intronic region (non-coding RNA)
	X	q22.2	103,186,126	103,353,105	13	gain	0.596946	8.82 × 10^−23^	*MIR1256*/-	*TMSB15B, H2BFXP, H2BFWT, H2BFM, SLC25A53*	
14.1	2	p23.3	25,875,529	25,930,894	6	loss	−0.727414	2.15 × 10^−11^	-	*DTNB*	Encompasses the first exon of the main isoform
26.1	1	q31.1	186,322,925	186,565,261	17	gain	0.877082	2.90 × 10^−64^	*MIR548F1*/-	*TPR, ODR4, OCLM, PDC*	Fully covers all genes, except the miRNA (1/2 exons)
27.1	5	p15.2	12,592,363	12,669,790	4	loss	−0.940583	1.18 × 10^−11^	-/*LINC01194*	-	Only intronic region
27.2	5	p15.2	12,592,363	12,669,790	4	loss	−1.072796	3.07 × 10^−17^	-/*LINC01194*	-	Only intronic region
	11	q13.2	66,831,965	66,984,794	13	gain	0.86627	1.79 × 10^−25^	-	*RHOD, KDM2A*	Encompasses 4/5 exons of *RHOD* and 8/21 exons of *KDM2A*
51.1	2	p25.1	9,802,238	9,886,004	4	gain	0.957729	1.59 × 10^−10^	-	-	
	19	p13.2	12,766,741	12,814,116	6	gain	1.046386	4.80 × 10-^11^	-	*MAN2B1, WDR83, WDR83OS, DHPS, FBXW9, TNPO2*	Fully covers WDR83, WDR83OS, DHPS, FBXW9, 13/24 exons of MAN2B1, and 6/26 exons of TNPO2
53	X	p22.33	1,731,671	1,746,762	4	gain	0.750151	1.12 × 10^−15^	-	*ASMT*	Encompasses 4/10 exons
58.1	22	q13.31-q13.32	46,613,498	48,444,626	156	gain	0.394413	2.94 × 10^−61^	-/*GTSE1,KLF3-AS1*	*PPARA, CDPF1, PKDREJ, TTC38, TRMU, CELSR1, GRAMD4, CERK, TBC1D22A*	
66	4	q25	110,754,346	110,812,252	5	loss	−0.713604	3.13 × 10^−10^	-	*RRH, LRIT3*	Encompasses 5/7 exons of *RRH* and the whole *LRIT3* gene
	7	q21.13	89,865,734	89,917,070	6	loss	−0.66345	1.47 × 10^−10^	-	*STEAP2, CFAP69*	Encompasses 1/6 exons of *STEAP2* and 14/23 exons of *CFAP69*
84.1	17	p12	14,100,118	15,442,066	68	loss	−0.794377	4.01 × 10^−102^	-/*MGC12916*, *CDRT7*	*COX10, CDRT15, HS3ST3B1, PMP22, TEKT3, CDRT4, TVP23C-CDRT4, TVP23C*	Fully covers all genes, except *COX10* (1/6 exons), *TVP23C* 1/6 exons), and *TVP23C-CDRT4* (2/7 exons)
	20	q13.12	44,125,965	44,408,171	25	gain	0.518682	4.41 × 10^−23^	-	*SPINT3, WFDC6, SPINLW1-WFDC6, SPINLW1, WFDC8, WFDC9, WFDC10A, WFDC11, WFDC10B, WFDC13, SPINT4, WFDC3*	Encompasses all of the genes, except *WFDC3* (3/7 exons)
162	8	q21.13	81,838,414	82,051,934	19	gain	0.559769	7.35 × 10^−23^	-	*PAG1*	
207	2	q11.2	101,488,408	101,727,546	22	gain	0.709334	2.37 × 10^−40^	-	*NPAS2, RPL31, TBC1D8*	
	4	p14	38,637,640	40,908,108	178	gain	0.397618	9.56 × 10^−123^	*MIR574*/*KLF3-AS1*	*KLF3, TLR10, TLR1, TLR6, FAM114A1, TMEM156, KLHL5, WDR19, RFC1, KLB, RPL9, LIAS, LOC401127, UGDH, SMIM14, UBE2K, PDS5A, LOC344967, N4BP2, RHOH, CHRNA9, RBM47, NSUN7, APBB2*	
229	12	q14.1	61,562,860	61,638,882	4	gain	0.720912	2.17 × 10^−10^	-		
	19	p13.3	4,809,281	4,889,734	10	loss	−0.619179	7.88 × 10^−13^	-	*TICAM1, PLIN3*	Fully covers both genes
339	1	p13.3	109,351,734	109,932,682	48	gain	0.396988	6.11 × 10^−35^	-/*SCARNA2*	*STXBP3, AKNAD1, LOC642864, GPSM2, CLCC1, WDR47, TAF13, TMEM167B, CFAP276, ELAPOR13, SARS1, CELSR2, PSRC1, MYBPHL, SORT1*	

Chr: chromosome.

## Data Availability

The original contributions presented in the study are included in the article/Appendix A, further inquiries can be directed to the corresponding author.

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
