# Peer review of "Epidemiological, Clinical, and Genomic Profile in Head and Neck Cancer Patients and Their Families"

_biomedicines, 2022, doi:10.3390/biomedicines10123278_

Round 1

Reviewer 1 Report

This is a well-designed study that builds on previous reports to give further evidence of the role of genetic predisposition for HNC. A strong point here is the coverage of development of other cancers as well as HNC and the work on germline copy variations.

The authors clearly highlight the limitations of the study in terms of numbers used, but the methodology used here gives a strong platform for more extensive studies in larger cohorts.

Author Response

We appreciate the positive comments and the analysis of our manuscript.

Reviewer 2 Report

The submitted manuscript describes preliminary findings about the familial cancer history, lifestyle, and other predisposing factors associated with head and neck cancer in the Brazilian population and the rare germline CNVs detected in some cases. I thought that the topic is interesting and falls within the scope of biomedicines. I have only a small suggestion.

・ As the authors noted, squamous cell carcinoma arising in the oropharynx, including the base of the tongue, is frequently associated with high-risk HPV. However, the detailed information about HPV in the oropharyngeal and oral cancers examined in this study is not clarified. I recommend that the HPV or p16 status of examined cases will be provided.

Author Response

As suggested, we included the HPV status of our cases. Among our dataset of 74 patients, five tumor samples (6.7%) were genotyped as HPV16 (one oral cavity and four oropharyngeal carcinomas). Two patients (tonsil and base of tongue carcinomas) investigated by array CGH were HPV16. The manuscript was modified accordingly (Materials and Methods, Results, Discussion and References sections, and Tables 1 and S1).